# Carbon Ion Radiotherapy: An Evidence-Based Review and Summary Recommendations of Clinical Outcomes for Skull-Base Chordomas and Chondrosarcomas

**DOI:** 10.3390/cancers15205021

**Published:** 2023-10-17

**Authors:** Adam L. Holtzman, Katharina Seidensaal, Alberto Iannalfi, Kyung Hwan Kim, Masashi Koto, Wan-Chin Yang, Cheng-Ying Shiau, Anita Mahajan, Safia K. Ahmed, Daniel M. Trifiletti, Jennifer L. Peterson, Daniel M. Koffler, Laura A. Vallow, Bradford S. Hoppe, Michael S. Rutenberg

**Affiliations:** 1Department of Radiation Oncology, Mayo Clinic, Jacksonville, FL 32224, USA; 2Department of Radiation Oncology, University Hospital Heidelberg, 69120 Heidelberg, Germany; 3Radiation Oncology Clinical Department, National Center for Oncological Hadrontherapy (CNAO), 27100 Pavia, Italy; 4Department of Radiation Oncology, Yonsei Cancer Center, Heavy Ion Therapy Research Institute, Yonsei University College of Medicine, Seoul 03722, Republic of Korea; 5QST Hospital, National Institutes for Quantum Science and Technology, Chiba 263-8555, Japan; 6Department of Heavy Particles & Radiation Oncology, Taipei Veterans General Hospital, Taipei City 11217, Taiwan; 7Department of Radiation Oncology, Mayo Clinic, Rochester, MN 55905, USA; 8Department of Radiation Oncology, Mayo Clinic, Scottsdale, AZ 85259, USA

**Keywords:** carbon ion radiotherapy, radiation therapy, cancer outcomes, base of skull, particle therapy, chordoma, chondrosarcoma

## Abstract

**Simple Summary:**

Curative-intent definitive treatment for chordoma and chondrosarcoma includes surgery, radiotherapy, or both. In patients with high-volume tumors, unresectable tumors, or recurrent disease, carbon ion radiotherapy has been used to reduce both relapse and treatment sequelae. Because of the limited number of carbon centers worldwide, there is difficulty in accumulating data and providing consensus guidance for its clinical use. Herein, we describe the clinical evidence to date and provide both a graded review of the available evidence and international summary recommendations for the indications of carbon ion radiotherapy in skull-base chordoma and chondrosarcoma.

**Abstract:**

Skull-base chordoma and chondrosarcoma are rare radioresistant tumors treated with surgical resection and/or radiotherapy. Because of the established dosimetric and biological benefits of heavy particle therapy, we performed a systematic and evidence-based review of the clinical outcomes of patients with skull-base chordoma and chondrosarcoma treated with carbon ion radiotherapy (CIRT). A literature review was performed using a MEDLINE search of all articles to date. We identified 227 studies as appropriate for review, and 24 were ultimately included. The published data illustrate that CIRT provides benchmark disease control outcomes for skull-base chordoma and chondrosarcoma, respectively, with acceptable toxicity. CIRT is an advanced treatment technique that may provide not only dosimetric benefits over conventional photon therapy but also biologic intensification to overcome mechanisms of radioresistance. Ongoing research is needed to define the magnitude of benefit, patient selection, and cost-effectiveness of CIRT compared to other forms of radiotherapy.

## 1. Introduction

Chordoma and chondrosarcoma are rare extradural sarcomas that can affect bones of the base of the skull. Each has distinct disease courses, of which treatment includes maximal safe resection and/or radiotherapy (RT) to reduce the risk of either recurrence or disease progression. Anatomically, regarding the skull base, while chordomas occur along the central neural axis, chondrosarcomas are typically lateralized, arising along the paramedian endochondral synchondrosis. For patients with chondrosarcoma, RT has been shown to decrease the risk of recurrence after maximal safe resection, and local control outcomes at 10 years following treatment are typically above 90%. Chordomas, on the other hand, tend to have a more insidious natural history with a bimodal recurrence distribution within the first few years after treatment, with a second peak in nearly 50% of patients 10 years after treatment despite adjuvant therapies [1,2].

The most common method for definitive or adjuvant RT delivery is external beam RT, which is a noninvasive method for delivering radiation to the treatment area. Historically, this consisted of X-rays or gamma-rays delivered with a linear accelerator or Cobalt-60 [3]. Due to poor disease control outcomes with conventional techniques and toxicities, other methods for radiation delivery have been explored to improve efficacy and reduce treatment morbidity. Of these, particle therapy (PT) has become a commonplace and promising strategy [2].

Because of the well-documented dosimetric properties of PT, treatments such as proton beam therapy (PBT) have been able to deliver high doses of radiation and spare critical nontargeted normal tissue from low to moderate doses of collateral radiation [1,2]. In fact, studies have shown that both doses of 70 Gy radiobiologic effectiveness (RBE) or higher and the use of PBT are associated with increased survival over conventional RT for these histologies [2,4]. PBT and other forms of highly conformal radiotherapeutic techniques, including X-ray delivery such as intensity-modulated RT (IMRT) and stereotactic radiosurgery (SRS), have enabled dose escalation with avoidance of sensitive adjacent neurovascular tissue near the target or tumor bed. However, both proton and photon modalities are limited in the potential to biologically intensify therapy beyond physical dose deposition alone. While ongoing studies regarding linear energy transfer (LET) optimization with PBT are underway, both protons and photons are considered low LET [5].

This has led to a great interest in the utilization of a specific subset of PT, heavy ion therapy, and, in particular, carbon ion RT (CIRT) [1,2,4,6,7,8,9]. Carbon has one of the more favorable entrances to peak dose ratios and distal fragmentation among the heavy ions. Therefore, the estimated RBE of CIRT, which ranges from 2.5 to 5, may be better suited for overcoming the mechanisms of radioresistance compared to low LET modalities [8,10]. In fact, in vitro studies have shown that both chordoma and chondrosarcoma are more sensitive to carbon ions compared to either X-rays or protons [11,12]. Thus, CIRT has the potential to both limit dose to surrounding normal tissue as with other techniques of highly conformal RT and also to biologically enhance the effects of RT. Because of the limited number and geographic distribution of carbon facilities worldwide, the number of patients treated with CIRT is limited, and the understanding of the best uses of this therapy is not well described [10].

Therefore, we sought to perform an evidence-based systematic review of the available evidence to date to provide a summary recommendation regarding the use of CIRT for skull-base chordoma and chondrosarcoma.

## 2. Materials and Methods

A comprehensive literature review of clinical outcomes was performed using Medline. Search terms included the following: “Chordoma”[MeSH Terms] OR “Chondrosarcoma”[MeSH Terms] AND “Heavy Ion Radiotherapy”[MeSH Terms]. The search terms included “Heavy Ion Radiotherapy”, as this was defined as “The use of a heavy ion particle beam for radiotherapy, such as the HEAVY IONS of CARBON”, by the National Library of Medicine, ensuring that a broad initial search criteria would capture all potentially eligible published studies. Additionally, references for selected studies were reviewed, and any publications not included in the search were manually added and reviewed in the screening. In total, 227 peer-reviewed articles written in English were extracted and included for evaluation. We followed the Preferred Reporting Items for Systematic Reviews and Meta-Analyses (PRISMA) guidelines, and the protocol has not been registered. The principal and senior authors initially screened the abstracts of these studies using an abstract review application, Covidence (Melbourne, Australia), which led to exclusion of 183 articles. As this review focused on grading the evidence of available clinical outcomes, reasons for exclusion included prohibitively small series, studies of experimental systemic agents, non-included pathologies or disease sites, incomplete radiation description, studies with significant heterogeneity or imbalance, and those reporting primarily technical or basic research. Twenty-four publications that report outcomes, toxicity, or relevant clinical information after carbon therapy for skull-base chordoma or chondrosarcoma were identified, as depicted in the PRISMA flow diagram in Figure 1. Appendix A summarize the key findings ordered by level of evidence in reverse chronological order separated by histologic diagnosis. The level of evidence was graded per the Oxford Centre for Evidence-Based Medicine, “The Oxford 2011 Levels of Evidence”, using an online iterative approach [13]. All studies included were then peer reviewed by all authors who reviewed the manuscript for those inclusions and level of evidence grading. This paper was circulated among co-authors for peer review as published in other consensus recommendations and international statements [14]. Among the studies included are 2 systematic reviews, 2 cost-effectiveness analyses, and 6 prospective analyses of carbon therapy for skull-base chordoma and chondrosarcomas from the German Ion Research Center (GSI; Heidelberg), the National Institute of Radiological Sciences (NIRS) in Japan, and the National Center for Oncological Hadrontherapy (CNAO) in Italy. Altogether, these institutions have enrolled more than 200 unique patients on prospective trials for skull-base chordoma and chondrosarcoma, and just as many patients have been analyzed retrospectively. Most patients received total doses around 60 GyE in 16 or 20 fractions with a typical treatment plan shown in Figure 2.

## 3. Results

### 3.1. Skull-Base Chordoma

In 2007, Schulz-Ertner et al. published an update on GSI’s phase 1/2 trials using active raster-scanning carbon therapy [15]. The goal of the trials was to determine treatment efficacy and toxicity. Among 96 patients with skull-base chordomas (44 treated on-trial and the rest treated per protocol) who received a median total dose of 60 GyE (range, 60–70 GyE) over 20 fractions and were followed for a mean 31 months (range, 3–91), the 3- and 5-year local control and overall survival rates were 80.6% and 70.0%, and 91.8% and 88.5%, respectively. Three patients (3%) experienced acute grade 3 mucositis. Late toxicities of grade 3 or higher included four patients with grade 3 optic nerve neuropathy and one with grade 3 necrosis of a fat graft. Thirteen patients experienced an improvement in tumor-related cranial nerve deficits present prior to carbon therapy, while two experienced improvements in motor and sensory deficits, two saw improvement in their ataxia, and three experienced pain relief.

In 2009, Mizoe et al. published their experience delivering 34 courses of CIRT in 33 patients (one of whom was treated twice) with skull-base chordomas under three prospective studies: a pilot study (4 treatments), a phase 1/2 dose-escalation study (16 treatments), and a phase 2 study (14 treatments) [16]. Patients received either 48.0 GyE (4 patients), 52.8 GyE (3 patients), 57.6 GyE (7 patients), or 60.8 GyE (19 patients) over 16 fractions. With a mean follow-up of 53 months (range, 8–129 months), they reported 5- and 10-year local control rates of 85.1% and 63.8% and overall survival rates of 87.7% and 67%, respectively. No acute or late grade 3 toxicities were observed. The multivariate analysis revealed that target doses exceeding 60 GyE were associated with higher local control rates.

Uhl et al. of GSI published outcomes after raster-scanning carbon therapy in 2014 [17]. Among 155 patients with chordoma treated to a total of 75 GyE and followed for a median 72 (range, 12–165) months, 3-, 5-, and 10-year local control rates were 82%, 72%, and 54%, and overall survival rates were 95%, 85%, and 75%, respectively [17]. The Heidelberg experience was updated in 2023, which included 111 patients treated with CIRT to a total dose of 66 GyRBE in 22 treatments [18]. With a median follow-up of 49 months, the 3- and 5-year local control and overall survival were 80% and 65%, and 91% and 83%, respectively. They performed a comparison of 36 patients treated with PBT and did not find any statistical difference in outcomes.

Two other recent retrospective studies have analyzed outcomes in chordoma patients. In 2018, Takagi et al. reported the experience of the Hyogo Ion Beam Medical Center using PT for chordoma, in which 13 patients were treated with carbons to doses between 57.6 GyE and 74.0 Gy over 16 to 37 fractions [19]. After a median follow-up of 56 months, the local control, progression-free survival (PFS), and overall survival rates at 5 and 8 years were 85% and 71%, 81% and 65%, and 86% and 76%, respectively. No acute grade 3 or higher toxicities were reported, but patients experienced late grade 3 or higher toxicities, including grade 3 brain necrosis (*n* = 2), optic nerve disorder (*n* = 1), nervous system disorders (*n* = 2), and middle ear inflammation (*n* = 1); moreover, one patient had a grade 4 pharyngeal hemorrhage. Most recently, in 2020, Koto et al. reported similar outcomes in a retrospective analysis of 34 prospectively enrolled patients with chordoma who received 60 Gy over 16 fractions at NIRS [20]. With a median follow-up of 108 months (range, 9–175 months), the 5- and 9-year local control rates were 76.9% and 69.2%, PFS rates were 65.7% and 53.8%, and overall survival rates were 93.5% and 77.4%, respectively. One patient had acute grade 3 mucositis, while serious late toxicities included grade 3 mucosal ulcer (*n* = 1), grade 4 ipsilateral optic nerve injuries (*n* = 2), and grade 5 mucosal ulcer (*n* = 1).

In 2020, Iannalfi et al. reported outcomes at CNAO after two prospective phase 2 trials delivering PT—either proton therapy or carbon therapy—for skull-base chordoma with identical inclusion criteria to determine treatment efficacy and toxicity [21]. Of 135 enrolled participants, 65 received carbon therapy to a total dose of 70.4 GyE over 16 fractions. At a median follow-up of 49 (range, 6–87) months, the 3- and 5-year local control rates were 77% and 71%, PFS rates were 87% and 84%, and overall survival rates were 90% and 82% for the patients treated with carbon therapy. Although toxicities were not delineated by PT type, no acute toxicities of grade 3 or higher occurred, and in terms of late effects, there was one grade 4 ototoxicity, two grade 4 eye injuries, and three grade 3 nervous system disorders. In all cases, the involved organs at risk were treated with therapeutic doses without compromising patients’ consent, and efforts were made to spare the contralateral organs at risk.

The highest level of evidence available is from a systematic review and meta-analysis performed by Lu et al. in 2020 [22]. They reported the local control incidence at 1-, 5-, and 10 years were 99%, 80%, and 56%, and the estimates of overall survival probability at 1-, 5-, and 10 years were 100%, 94%, and 78%, respectively. The incidence of early and late toxicity (Grade  ≥  3) ranged from 0% to 4% across all study groups.

### 3.2. Skull-Base Chondrosarcoma

In 2007, Schulz-Ertner et al. reported that of 54 patients treated for skull-base chondrosarcoma, 37% had disease of the sphenopetrosal synchondrosis and 33.3% of the parasellar region [23]. After a median dose of 60 GyE (range, 57–70 GyE) over 15 fractions, and a mean follow-up of 33 months (range, 3–84 months), the 3- and 4-year local control rates were 96.2% and 89.8%, respectively, and the overall survival rates were both 98.2%. Only two grade 3 or higher toxicities were reported, with one patient developing acute grade 3 mucositis and one developing grade 3 abducent nerve paresis. Following carbon therapy, four patients experienced an improvement in pre-existing cranial nerve deficits (abducent nerve paresis, 2 patients; oculomotor nerve paresis, 1 patient; facial nerve paresis, 1 patient).

Uhl et al. of GSI published their outcomes after raster-scanning carbon therapy for chondrosarcoma in 2014 [24]. Among 79 patients with chondrosarcoma treated with a median total dose of 60 GyE (range, 57–69 GyE) and with a median follow-up of 91 (range, 3–175) months, 3-, 5-, and 10-year local control rates were 95.9%, 88%, and 88%, respectively. The overall survival rates at the same timepoints were 95%, 85%, and 75%, respectively. Toxicities were not graded.

In a 2018 update from GSI, Mattke et al. reported the combined proton and carbon experience treating chondrosarcoma [25]. Among 101 patients, 79 of whom were treated with carbon therapy and followed for a median of 43.7 months, the 1-, 2-, and 4-year local control rates after carbon therapy were slightly higher than previously reported, at 98.6%, 97.2%, and 90.5%, respectively, as were the overall survival rates, at 100%, 98.5%, and 92.9%.

Within the aforementioned review by Lu et al. [22], the local control incidence at 1-, 5-, and 10 years in chondrosarcoma-only studies were 99%, 89%, and 88%, respectively, and the estimates of overall survival probability were 99%, 95%, and 79%.

### 3.3. Combined Outcomes

In summary, there are nine experiences that combined skull-base chordoma and chondrosarcomas, including over 250 patients [26,27,28,29,30,31,32,33]. Overall, median total doses ranged from 48 to 80 GyE, and local control was reported over 90% at 2 years among the series. The highest level of evidence, however, comes from a systematic review completed by Dong et al. in 2022 [34]. Of the nearly 900 patients with skull-base sarcomas who received CIRT, 526 had chordoma and 255 had chondrosarcoma. The local control rate at 5 and 10 years in these studies were 74.3% (95% CI = 0.666–0.820, I 2 = 85.2%) and 64.7% (95% CI = 0.451–0.843, I 2 = 95.3%), and the overall survival was 72.7% (95% CI = 0.609–0.844, I 2 = 95.3%) and 72.1% (95% CI = 0.661–0.781, I 2 = 46.5%), respectively. Across all studies, the incidence of acute grade 3 toxicity was 3.2% to 3.8%, and late grade 4 was 2.1% to 8.0%.

### 3.4. Re-Irradiation

While several of the previously mentioned series included small cohorts of patients who received prior radiation [16,32,33,35], there are three reported outcome studies from GSI that exclusively assess longitudinal outcomes following re-irradiation with CIRT [31,36,37]. The first two initial reports in 2011 evaluated the re-irradiation of acute toxicities in head and neck tumors [36] and those of the brain, skull base, and sacrum [37]. Jensen et al. found that four of the five patients with chordoma or chondrosarcoma achieved stable disease after re-irradiation, and there were no acute grade 3 toxicities seen [36]. Combs et al. reported on 16 patients with chordoma and 2 with chondrosarcoma of the skull base, who all tolerated re-irradiation without interruptions [37]. While this study was not histologically specific for the reported outcomes, for skull-base tumors, local control after re-irradiation was 64% at 3 years, and overall survival was 43% at 5 years.

Uhl et al. published a third series from GSI of outcomes after raster-scanning carbon [31]. Among the 25 patients with either chordoma or chondrosarcoma who were re-irradiated with a total dose of 51 (45–60) or 63.8 Gy (56.2–75), the 2-year probability of local PFS was 79.3%. One patient developed grade 3 osteoradionecrosis.

### 3.5. Pediatrics

As pediatric patients constitute less than 5% of all new chordoma and chondrosarcoma diagnoses, there is only one known pediatric series to date. In 2009, Combs et al. published their findings from a prospective analysis of treatment efficacy and toxicity in 17 pediatric patients with skull-base tumors (chordoma, 7; chondrosarcoma, 10) treated with carbon therapy [29]. The median age was 17 years (range: 5–21 years), and 10 were over 18 years old. With a median follow-up of 49 (range: 3–112) months, the local control rate was 94%. The authors also reported adrenocorticotropic hormone and growth hormone deficiency in one patient and gonadotropin deficiency in another.

### 3.6. Cost Effectiveness

Two series have evaluated the cost-effectiveness of CIRT with skull-base chordoma, while none have evaluated this for chondrosarcoma. Jäkel et al. noted that if the local control rate for skull-base chordoma achieved with CIRT exceeds 70%, the overall treatment costs for CIRT are lower than for conventional RT. The cost-effectiveness ratio for CIRT compared to photon RT was EUR 2539 per 1% increase in survival or EUR 7692 per additional life year; however, the study did not evaluate differences compared to proton RT [38].

In 2018, Sprave et al. conducted a similar analysis using 10-year outcomes. They found that the quality-adjusted life-year (QALYs) outcomes were 6.65 for photon RT and 8.26 for CIRT, a difference of 1.61 discounted lifetime QALYs for patients treated with CIRT [39]. The overall incremental cost-effectiveness ratio was EUR 8855.76/QALY, suggesting that CIRT is a highly cost-effective treatment option for chordoma.

## 4. Discussion

In summary, the reported evidence, as reviewed and outlined in Appendix A, shows favorable local control, overall survival, and toxicity among the reported series following CIRT for both skull-base chordoma and chondrosarcoma. The highest quality of evidence estimates of the 5 and 10-year local control for chordoma and chondrosarcoma following CIRT is 80% and 56%, and 89% and 88%, respectively [22].

### 4.1. Comparisons to Other Forms of Radiotherapy

Highly conformal external beam techniques for radiation delivery to skull-base tumors are widely available worldwide. Nearly all modern linear accelerators can deliver IMRT or SRS. These are treatments delivered with X-rays or photon-based RT. In fact, Sahgal et al. reported on 42 patients treated with image-guided IMRT, with either skull-base chordoma (*n* = 24) or chondrosarcoma (*n* = 18). With a median follow-up of 36 months, 5-year overall survival and local control rates were 85.6% and 65.3% for patients with chordoma and 87.8% and 88.1% for those with chondrosarcoma, respectively. This series demonstrated the safe delivery of a median dose of 70 Gy for chondrosarcoma and 76 Gy for chordoma while reporting eight late radiation-induced toxicities. Despite treating the brainstem and optics up to a maximum of over 70 Gy for certain circumstances, they did not note any brainstem necrosis or radiation-induced optic neuropathy, although they did observe one grade 5 radiation-induced secondary malignancy.

SRS has also been used for the postoperative and salvage management of chordomas. One series reported 5-year PFS as 43% [40], which was similar to the reported literature ranging from around 20% to as high as 80% [40,41,42,43,44,45]. Radiosurgery has been particularly useful in cases of tumor recurrence following prior irradiation. The group at MD Anderson Cancer Center evaluated 40 episodes of either local or distant progression treated in 16 patients with skull-base chordoma [45]. While the median disease-specific survival was only 36 (range: 2–114) months, in multivariate analysis, SRS was the only treatment modality associated with improved freedom from treatment site progression.

PBT is a form of specialized PT, which, while still geographically limited, has become more accessible over the last decade for certain countries and indications [46,47]. Some advantages of PBT compared to photon-based treatment options such as IMRT or SRS are the reduction in integral dose and the low to moderate dose bath, which have been shown to decrease the risk of long-term treatment complications, including secondary malignancy [48]. Reports also show a low incidence of radiation optic neuropathy [49] and brainstem injury [50] and high efficacy with PBT, particularly with dose-escalated therapy [1,2,4,6,7,51].

While several institutions have reported on experiences of patient treatment with both proton and carbon, there are no randomized comparisons published to date [18,19,21,25,52]. However, two phase III studies are currently enrolling evaluating CIRT compared to PBT for skull-base chordoma (NCT01182779) and chondrosarcoma (NCT01182753) [10].

### 4.2. Recommendations for CIRT

While maximal safe surgery remains the primary treatment, because of the high likelihood of residual tumor, adjuvant PT is used to improve local control. Based on the quality of the present literature, the strength of the following recommendation for CIRT is an option because of the quality of the data and limited availability, as summarized in Table 1. Based on this evidence, CIRT should be considered for those with the following:Unresectable or subtotally removed chordoma or chondrosarcoma;High-volume residual disease (>25 mL);Recurrence following prior surgery;Recurrence following prior irradiation;Tumors not encroaching or directly abutting the brainstem or optic apparatus.

Data from ongoing phase II and III prospective randomized studies are still needed. Furthermore, to fully maximize the potential advantages of CIRT, better patient selection with personalized biomarkers may identify those with radioresistant cancers and, therefore, most likely to benefit from this therapy [10]. While other prior systematic reviews exist [22,34], this is the first to not only summarize the quality of clinical evidence but also provide a multi-institutional, recommendation based on the available data, which is valuable for treatment decision making and resource allocation because of the limited availability and access to CIRT.

### 4.3. Limitations

This study has several limitations as it is a systematic and evidence-based review that relies predominantly on retrospective studies, cohorts with overlapping patient populations, limited prospective interventions, and no randomized clinical trials. Additionally, because of the narrow geographic scope of the operating carbon centers worldwide, they may not accurately represent the global diversity of the disease, potentially leading to skewed treatment outcomes and selection biases in the patients who have the physical capacity and socioeconomic resources to travel for therapy. Patients from underrepresented regions may have unique characteristics or access to different healthcare resources that are not adequately captured in the review, rendering the findings less generalizable. Moreover, the concentration of research within a few centers can introduce institutional and regional biases and variations in clinical practices, all of which can impact the overall quality and reliability of the evidence. Lastly, the reporting and heterogeneity of treatment technique, plan optimization methods, and number of beam and treatment specifics have been inconsistent across the literature. Therefore, when interpreting the results, it is essential to acknowledge these constraints as yet another layer of potential bias that must be considered when comparing treatment options and the strength of the consensus-based recommendations.

## 5. Conclusions

CIRT allows for both improved physical dosimetry and enhanced biological effects with high LET particles over conventional photon RT. CIRT results in excellent clinical outcomes with an acceptable toxicity profile for the treatment of skull-base chordoma and chondrosarcoma and should be considered a treatment option for cases in which radiosensitivity of nearby normal tissues and/or radioresistant tumor histology is of concern.

## Figures and Tables

**Figure 1 cancers-15-05021-f001:**
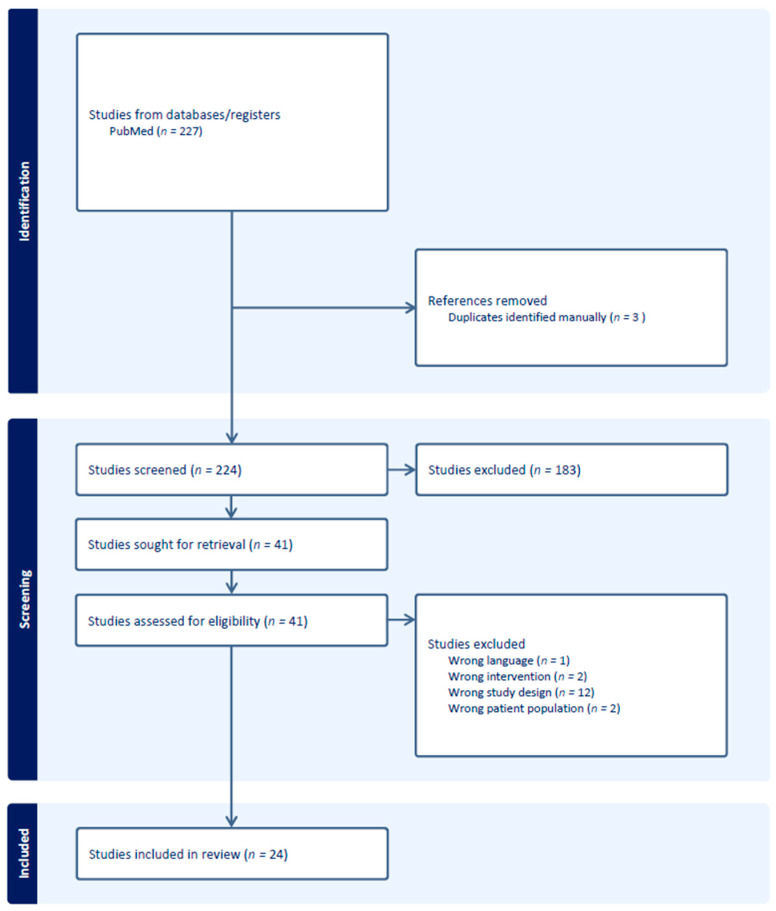
Search results per PRISMA guidelines.

**Figure 2 cancers-15-05021-f002:**
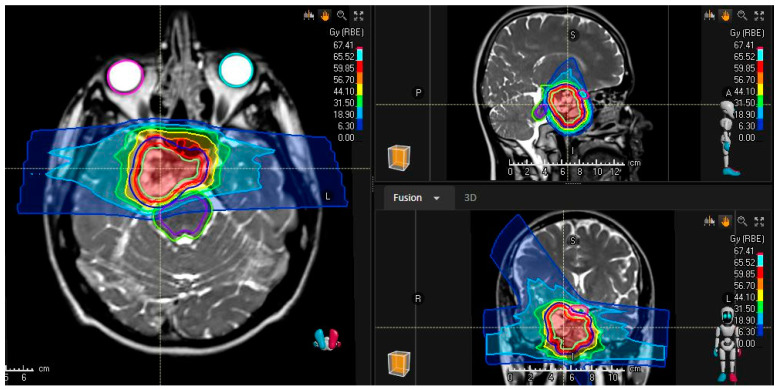
Representative T2-weighted magnetic resonance image co-registered to the treatment planning scan illustrating the color wash dose distribution of a 3-field technique using carbon ion radiotherapy in a patient with a low-grade, petroclival chondrosarcoma being treated to 63 Gy radiobiologic effectiveness in 21 fractions with a horizontal beam line.

**Table 1 cancers-15-05021-t001:** Evidence-based recommendation: CIRT for skull-base chordoma.

Items	Explanation
Aggregate Grade of Evidence	Chordoma: C (Level 2: 1 study; Level 3: 3 studies; Level 4: 6 studies) Chondrosarcoma: C (Level 2: 1 study; Level 3: 1 study; Level 4: 2 studies) Combined series: (Level 2: 1 study; Level 3: 2 studies; Level 4: 6 studies)
Benefit	CIRT provides LC and OS benefits for skull-base chordoma and chondrosarcoma when used as monotherapy or adjuvantly following surgery.
Harm	CIRT morbidity is related to the extent and site of the tumor and toxicities are in line with reported literature compared to other highly conformal radiation delivery techniques.
Cost	Because of the increased tumor control in published models for chordoma, CIRT is reported to be highly cost-effective. There are no such analyses available for chondrosarcoma.
Benefit-Harm Assessment	Preponderance of benefits over harms.
Value Judgments	CIRT should be considered for radioresistant histologies with gross residual disease at the time of treatment or following treatment recurrence.
Policy Level	Option
Intervention	CIRT should be considered for improving LC and OS when weighed for patient-specific and tumor features and is optional when available.

Abbreviations: CIRT, carbon ion radiotherapy; LC, local control; OS, overall survival.

## Data Availability

Not applicable.

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
