# Peer review of "Carbon Ion Radiotherapy: An Evidence-Based Review and Summary Recommendations of Clinical Outcomes for Skull-Base Chordomas and Chondrosarcomas"

_cancers, 2023, doi:10.3390/cancers15205021_

Round 1
Reviewer 1 Report
Dear Editor
The review paper can be accepted after minor revision in cancers journal. But some items should be addressed:
1- why did the authors used only 36 ref. for review paper which is not enough?
2- the introduction section needed to be improved. An in depth explanation of the RT, specifically
3- what is the benefits of carbon therapy in comparison with other kinds of RT?
In general, i believe that the paper should be improved completely in all sections
Reviewer 2 Report
This is an interesting paper summarizing the published literature on this rare indications using carbon ion radiotherapy.
However the paper could be improved so it might be better readable and comprehensible.
Methods:
l78: any published studies not included.. This sounds as if there might be a systematic bias by including studies more or less on an undefined way. Please specify the method.
Figure 1: 183 of 224 screened studies were excluded. It is not described why these were excluded
Table 1: One might suspect that the reported outcomes from institutions with repeated publications include data from the same patients, possibly influencing the conclusions. This should be adressed in the methods and in the discussion. Which efforts were done to avoid weighting double publications.
Results: A repetition of the information already presented in the tables is unnecessary and makes the results hard to read. I would prefer to summarize and interprete the tables contents with regard to the parameters presented (local control/ OS / Tox).
The role of reviews in contrast to the reported patient studies with small patient numbers should be clarified and discussed later on.
Reviewer 3 Report
review comments:
The authors conducted a review regarding Carbon ion Radiotherapy for Skull Base Chordomas and Chondrosarcomas: An Evidence-Based Review and Summary Recommendations of Clinical Outcomes
*/Strengths:/*
— Well-established methods for systemic reviews used
— Clear tables in which most important summary of the studies can be found and interpreted easily
— Properly critical approach of the authors on the results
Even though the work was done with knowledge in the field, there are several issues to be pointed out:
*/Specific issues:/*
— the details of the carbon ion technique used in the reported studies (i.e., scattering or active scanning technique, planning optimization technique, number of beams used etc.) should be provided in a separate table;
— the wide limitations of the study should be better reported in the paper, and the comments regarding the results of the paper should be made accordingly.
— the currently ongoing trials in this field should be reported in a table and commented in the discussion.
Reviewer 4 Report
The review proposed by Holtzman et al. aims to present evidence of the benefit of carbon ion radiotherapy to treat skull base chordomas and chondrosarcomas, as well as some guidelines.
The review is based on the analysis of 227 studies from those 24 were included.
The authors seem to have followed the PRISMA Guidelines for their work. However, it is difficult to understand why they limited their research to chordoma OR chondrosarcoma OR heavy ions and why « carbon ion » was not selected. If the title relies on this radiotherapy, at least the term should have been applied in the process to select the studies; Plus, the timeline of the analysis and process to exclude studies is not clear. Did authors do a peer review to keep or exclude analysis?
The manuscript is difficult to read with lots of tables aligned without any explanations related to them.
English is fine
